# Gut–Brain Interactions and Their Impact on Astrocytes in the Context of Multiple Sclerosis and Beyond

**DOI:** 10.3390/cells13060497

**Published:** 2024-03-13

**Authors:** Julia Zißler, Veit Rothhammer, Mathias Linnerbauer

**Affiliations:** Department of Neurology, University Hospital Erlangen, Friedrich-Alexander University Erlangen-Nuremberg, 91054 Erlangen, Germany

**Keywords:** gut–brain interactions, multiple sclerosis, neuroinflammation, microbiome, dysbiosis

## Abstract

Multiple Sclerosis (MS) is a chronic autoimmune inflammatory disease of the central nervous system (CNS) that leads to physical and cognitive impairment in young adults. The increasing prevalence of MS underscores the critical need for innovative therapeutic approaches. Recent advances in neuroimmunology have highlighted the significant role of the gut microbiome in MS pathology, unveiling distinct alterations in patients’ gut microbiota. Dysbiosis not only impacts gut-intrinsic processes but also influences the production of bacterial metabolites and hormones, which can regulate processes in remote tissues, such as the CNS. Central to this paradigm is the gut–brain axis, a bidirectional communication network linking the gastrointestinal tract to the brain and spinal cord. Via specific routes, bacterial metabolites and hormones can influence CNS-resident cells and processes both directly and indirectly. Exploiting this axis, novel therapeutic interventions, including pro- and prebiotic treatments, have emerged as promising avenues with the aim of mitigating the severity of MS. This review delves into the complex interplay between the gut microbiome and the brain in the context of MS, summarizing current knowledge on the key signals of cross-organ crosstalk, routes of communication, and potential therapeutic relevance of the gut microbiome. Moreover, this review places particular emphasis on elucidating the influence of these interactions on astrocyte functions within the CNS, offering insights into their role in MS pathophysiology and potential therapeutic interventions.

## 1. Introduction

Multiple Sclerosis (MS) is an autoimmune inflammatory disease of the central nervous system (CNS) that causes extensive damage to gray and white matter [1]. Pathogenesis involves the transgression of autoreactive immune cells into the CNS through the blood–brain barrier (BBB), thereby triggering localized inflammatory processes. This inflammatory response culminates in extensive demyelination and axonal degeneration and the formation of distinctive lesions, which are typical hallmarks for MS diagnosis [1]. Although the specific mechanisms underlying the infiltration of autoreactive immune cells into the CNS have not been elucidated, various genetic and environmental risk factors have been identified that contribute to elevated susceptibility. Among these, the gut microbiome is gaining particular increasing attention because alterations in its composition may significantly contribute to the susceptibility and progression of MS.

However, comprehending the true significance of the gut microbiome in the context of MS requires a thorough exploration of the intricate and highly diverse pathogenesis of the disease. The most common form of MS is relapsing–remitting MS (RRMS), which is characterized by recurring acute relapses followed by periods of recovery [1]. Over time, the majority of patients diagnosed with RRMS progress to a more chronic form, secondary progressive MS (SPMS). SPMS is characterized by gradual worsening of symptoms without full recovery. A small percentage of patients are diagnosed with primary progressive MS (PPMS), which is defined by a continuous worsening of the disease from disease onset [2].

Localized inflammation occurs in all phases of MS but is most pronounced in the acute phase, when autoreactive immune cells migrate from the periphery into the CNS. Among these peripheral immune cells, Th17 cells play a particularly vital role in disease development [3]. While it is not entirely clear why these cells become autoreactive, some theories suggest that these autoreactive cells escape tolerance mechanisms in the thymus and are activated in the periphery by molecular mimicry, superantigens, or bystander activation [4]. Once in the CNS, they are reactivated by antigen-presenting cells (APCs), such as macrophages, microglia, B cells, and potentially astrocytes [1]. While the exact autoantigen has not yet been identified, it is clear that the reactivation of Th1 and Th17 cells in the CNS results in their secretion of proinflammatory cytokines such as IFN-γ and IL-17, respectively, leading to monocyte recruitment and the activation of macrophages and microglia [1].

The presence of these activated innate and adaptive immune cells in the CNS consequently culminates in extensive demyelination and axonal degeneration, leading to the cognitive deficits and motor symptoms often observed in patients during relapse [5]. Moreover, the reactivation of autoreactive immune cells by APCs initiates a cascade that further fuels the inflammatory environment and hinders regenerative processes. While both microglia and macrophages are considered “janitors of the CNS” and possess the capacity to clear dead cells and debris, they fail to fulfill their beneficial functions in MS, thereby promoting demyelination and neurodegeneration [6]. Activated microglia are also able to polarize astrocytes to a proinflammatory activation state, thereby suppressing tissue-protective functions and fostering the infiltration of peripheral immune cells [7]. In addition to T cells, macrophages, and microglia, B cells have emerged as key cellular players in the pathogenesis of MS. The growing interest in their functions during the acute stages of MS has undoubtedly been fueled by the recent therapeutic success of B cell-targeting therapies. Although their exact contribution to demyelination is not fully understood, the presence of oligoclonal bands (OCBs) secreted by plasma cells is a defining feature and diagnostic criterion of MS. Altogether, the dense infiltration of immune cells into the CNS and the secondary initiation of CNS-intrinsic processes drive the formation of destructive lesions in the acute stages of MS. Infiltration and acute inflammation are also primary targets of the majority of disease-modifying therapies (DMTs), including Natalizumab (anti-alpha-4-Integrin), Rituximab (anti-CD20), or Fingolimod (S1PR antagonist), among others. Recently, Bruton tyrosine kinase (BTK) inhibitors have emerged as a new class of therapeutics for the treatment of RRMS. Here, the inhibition of BTKs modulates B cell responses without affecting their viability [8].

In progressive types of MS, inactive and chronic active lesions are more prominent [2,9]. However, in RRMS, white matter is predominantly affected, and demyelinated lesions occur increasingly frequently in gray matter as the disease progresses, explaining the increased decline in cognitive function in patients with progressive MS. Interestingly, lesions in gray matter show only a small percentage of infiltrated macrophages and T cells, suggesting the presence of processes beyond those mediated by infiltrating immune cells that define the acute stages of the disease [2,10].

In these lines, astrocytes in particular have gained increasing attention because they drive CNS-intrinsic inflammatory processes in chronic stages of this disease not only by the formation of glial scars, which represent a physical barrier that limits regeneration and remyelination, as well as impeding remyelination, e.g., via the production of chondroitin sulfate proteoglycans, lactosyl ceramide, tenascis, and hyaluronic acids [11,12,13,14]. While there is research suggesting that proinflammatory polarization of astrocytes induces neurotoxicity through their secretion of saturated lipids, there are also opposing data showing that reactive astrocytes also promote regeneration, suggesting a dual role in acute and chronic disease stages [15,16,17,18,19,20]. In addition to specific inflammatory stimuli that are provided by infiltrating and CNS-resident immune cells, exposure to gut-derived metabolites has recently been implicated in the regulation of astrocyte functions. The subsequent sections of this review delve into the intricate functions of the gut microbiome in the context of health and disease, with a specific focus on its relevance in MS and the pleiotropic functions of bacterial metabolites in facilitating cross-organ interactions.

## 2. The Gut Microbiome

The term microbiome describes a microecosystem in the body inhabited by microbes such as bacteria archaea, fungi, and viral entities. While many distinct microbiomes can be found in the various compartments of the body, the gut microbiome houses the largest number and diversity of microbes. The gut coevolved over thousands of years with a complex network and relationship between the microbiome and the host [21]. The composition of the gut microbiome can vary between individuals and is highly susceptible to environmental changes. The quantity and diversity of microbes differ depending on the location within the gastrointestinal tract [22]. However, approximately 12 different bacterial phyla were found to be the most predominant, including Firmicutes, Actinobacteria, Proteobacteria, and Bacteroidetes [23]. Although most of these phyla are commensal and nonharmful, they constitute a permanent threat of invasion for the host. To maintain homeostasis, the host’s strategy is to minimize contact between microbes and intestinal surfaces. This can be achieved by establishing physical mucus barriers, secreting enzymes and antimicrobial proteins, as well as immunological factors such as IgA that control the colonization of bacteria [24]. Altogether, these mechanisms are part of a bidirectional interaction between the host and the gut microbiome that is required for maintaining a “healthy” gut microbiome.

### 2.1. Functional Aspects of a “Healthy” Gut Microbiome

A major function of the gut microbiome lies in nutrient metabolism from the host’s diet, including carbohydrates, proteins, and lipids. These metabolic pathways yield essential amino acids and metabolites, e.g., through the fermentation of nondigestible dietary fibers to short-chain fatty acids such as acetate, propionate, and butyrate [25]. These SCFAs play a crucial role in maintaining homeostasis across various tissues and cell types. Notably, butyrate serves as a vital energy source for colonocytes, contributing significantly to gut health, especially considering that its precursor is the potentially toxic metabolic byproduct D-lactate [26,27]. Propionate, another SCFA, regulates gluconeogenesis and ATP production in the liver, while acetate plays a major role in cholesterol metabolism, lipogenesis, and appetite regulation [28].

In addition, the gut microbiome serves another crucial function by actively participating in the synthesis of both essential and nonessential amino acids. Moreover, the microbiome is responsible for the absorption of essential mineral nutrients such as magnesium, calcium, and iron [21,29]. Additionally, the microbiota produces certain vitamins, with various members of the gut microbiome capable of synthesizing vitamin K and most B vitamins, including biotin, cobalamin, folates, nicotinic acid, pantothenic acid, pyridoxine, riboflavin, and thiamine [21,29].

A “healthy” microbiome also constitutes an important line of defense against pathogens. For instance, certain bacterial phyla produce peptides with antimicrobial activity, called bacteriocins, which prevent the invasion of other species and may furthermore act as important signaling molecules for host–bacteria interactions and interspecies bacterial communication [30].

In conclusion, the gut microbiome plays a pivotal role in a diverse array of functions that reach far beyond metabolism and plays important roles in the host immune system. This symbiotic relationship between the gut microbiome and the host is characterized by mutual interdependence and underscores its significance as a major player in the context of autoimmune diseases.

### 2.2. The Role of the Gut Microbiome in Disease

The occurrence of immune-mediated, metabolic, and neurodegenerative diseases is increasing rapidly worldwide [31]. In the last few years, it has become increasingly evident that the gut microbiome contributes to disease development and pathogenesis in various contexts. The significance of the microbiome has been most convincingly demonstrated by transplant experiments in germfree (GF) mice, where reconstitution by certain bacteria was required for disease susceptibility [32,33].

In “healthy” individuals, the gut hosts a rich diversity of microbial species complemented by an intact intestinal barrier that serves as a crucial shield against bacteria and their mediators and forms a first line of defense by exerting a barrier effect called colonization resistance, which prevents exogenous microbes from proliferating in the gut microenvironment. [34] Given these multifaceted protective functions, any alteration in the composition of the gut microbiome may lead to harmful alterations, collectively referred to as dysbiosis, which typically involve the loss of beneficial bacteria, an overall reduction in microbial diversity, and the overgrowth of pathobionts [35]. Dysbiosis can be triggered by various factors, with dietary choices and genetics playing pivotal roles in shaping the microbiome composition [36]. Additionally, the inflammatory status of the host, especially during infections, can contribute to dysbiosis by compromising the colony resistance of commensals and ultimately impairing their ability to resist invading microbes [35].

As the immune system and gut microbiome are intricately intertwined, studying their interactions becomes crucial for understanding diseases affected by dysbiosis. These interactions are mediated by a multitude of microbial signaling molecules that engage a wide array of pattern-recognition receptors (PRRs) divided into four classes: C-type lectin receptors (CLRs), retinoic acid-inducible gene (RIG)-I-like and Toll-like (TLR) receptors (RLRs), and NOD-like receptors (NLRs) [37]. These receptors are expressed by both professional and nonprofessional antigen-presenting cells (APCs), and their signaling triggers the activation of various downstream proinflammatory cascades, including the expression of chemokines and cytokines. While CLR, TLR, and NLR primarily recognize microbial substances, RIG-I-like receptors are mainly responsible for viral sensing [37]. For instance, peptidoglycans produced by the microbiome are sensed via NOD2, which acts as an inducer of the nuclear factor kappa-light-chain-enhancer of activated B cells (NF-κB) and the inflammasome, resulting in the release of cytokine and antimicrobial products [37]. In addition to NLRs, TLRs are involved in the inflammatory response to pathogen-derived mediators. For instance, TLRs expressed on intestinal dendritic cells (DCs) sense microbial compounds, leading to the secretion of IL-23, stimulating mucosal innate immune defense and antimicrobial peptide production of innate lymphoid cells (ILCs) [38,39,40]. Equally important is the production of secretory IgA (sIgA) by B cells in Peyer’s patches, a process dependent on interaction with follicular T helper (T_FH_) cells, leading to the opsonization of bacteria [34]. Furthermore, IgA leads to the production of IL-10 by macrophages, resulting in an immunosuppressive state [41].

During dysbiosis, these responses are impaired, resulting in far-reaching effects on host defense and systemic immunity. The consequences of a disturbed microbiome are strongly dependent on individual effects on the bacteria colonizing the gut. For instance, some dysbiotic microbes, such as Porphyromonas gingivalis, activate NF-κB and thereby induce inflammatory signaling [42]. Furthermore, Sutterella and Pseudomonas species can degrade or inactivate sIgA, driving inflammation and the breakdown of the intestinal barrier [43]. Once dysbiosis is established, a harmful feedback loop is likely to develop between the host immune system and the microbiome, cross-regulating each other [35]. In recent years, numerous studies have linked the gut microbiome to various diseases, including autoimmune conditions such as Rheumatoid Arthritis (RA), MS, or primary neurodegenerative diseases such as Parkinson’s disease (PD) and Alzheimer’s disease (AD) [44,45,46,47], suggesting that the gut microbiome is a prime target for therapeutic intervention.

### 2.3. The Gut Microbiome in Patients with Multiple Sclerosis

The significance of the gut microbiome in MS is most strikingly supported by the fact that GF mice are protected from experimental autoimmune encephalomyelitis (EAE), a preclinical mouse model of MS [48,49]. Furthermore, GF mice that received stool samples from MS patients regained susceptibility, whereas mice that received stool samples from healthy twin donors remained insusceptible [50]. These findings emphasize the significance of the gut microbiome in MS and suggest that the associated dysbiosis contributes to disease development and progression.

In recent decades, an increasing number of studies have investigated the microbiome composition in MS patients, with compelling evidence demonstrating a strong association between dysbiosis, MS risk, disease course, and disease progression [51]. However, the microbial composition appears to vary based on disease stage and treatment. Additionally, the results of various studies exhibit discrepancies, underscoring the complexity of understanding the nuanced relationship between the microbiome and MS. In RRMS, multiple studies have reported a decrease in Bacteroidetes, especially Parabacteroides and Prevotella, while other genera, such as Psuedomonas, Mycoplana, Haemophilus, Blautia, and Dorea, were shown to increase [52]. Notably, patients on DMTs show an increased abundance of Prevotella and Sutterella and a decreased abundance of Sarcina compared to untreated patients [53], suggesting that therapeutic intervention also affects microbial composition. In progressive MS, distinctive alterations in microbiome composition, characterized by increases in Enterobacteriaceae, Streptococcaceae, and Clostridium and decreases in Agathobaculum and Blautia, have been observed [54].

In addition, multiple studies have demonstrated an increase in Akkermansia in the acute and chronic stages of MS [51,53,54]. Members of the Firmicutes phylum are considered significant contributors to MS, with evidence suggesting that high Firmicutes abundance is associated with increased relapse risk [55]. Notably, specific genera within Firmicutes, such as Blautia and Dorea, are increased in MS patients [51,52]. Interestingly, although Dorea is a typical constituent of a “healthy” gut, certain species demonstrate proinflammatory functions by inducing IFN-γ, indicating that Dorea can exert both pro- and anti-inflammatory effects through the surrounding microbiome [56].

Additionally, segmented filamentous bacteria (SFBs) have attracted considerable interest due to their potential association with autoimmune neuroinflammation. Colonization of the intestine with SFBs alone is sufficient to induce susceptibility to EAE and has been shown to promote Th17 cell production in the gut and CNS [57,58]. SFB colonization has also been linked to elevated expression of genes associated with inflammation and antimicrobial defense [57]. In addition to SFBs, certain strains of Clostridioides difficile have been demonstrated to activate immune cells. Specific strains carry surface layer protein A (SLPA), which exhibits cross-reactivity with myelin basic protein (MBP) 89–98, potentially leading to the activation of autoreactive MBP-specific T cells [59].

Notably, many alterations in the gut microbiome composition observed in MS patients are also evident in other autoimmune diseases [60,61,62], suggesting common underlying mechanisms that highlight the relevance of the gut microbiome in the context of autoimmunity. In summary, these findings indicate that the gut microbiome may be a significant contributor to the development and pathogenesis of MS, highlighting the intricate connection between the gut and the CNS.

## 3. The Gut–Brain Axis

The brain and gastrointestinal tract are intricately linked through both physical and biochemical pathways, constituting a bidirectional relationship commonly known as the gut–brain axis (GBA). It is worth noting that while the GBA is a widely used term, it is conceivable that the spinal cord is also impacted by these connections. In the following sections, this review categorizes the interaction between the gut and the CNS into distinct subsections: the different pathways of the GBA, the gut-derived signals involved in MS, how these signals affect reactive astrocyte functions, and, ultimately, their potential as therapeutic targets for the treatment of autoimmune neuroinflammation (Figure 1).

### 3.1. Vagus Nerve and Enteric Nervous System Interactions

A significant portion of the complexity in the interactions between the gut and the CNS stems from the absence of a direct communication route between these two organs. Instead, bidirectional interactions are facilitated by the enteric nervous system (ENS), which acts as a bridge that connects the cognitive centers of the CNS to the peripheral functions of the intestines (Figure 1), exhibiting structural similarities to the CNS [63]. In these lines, particularly enteric glial cells (EGCs), the most abundant cellular component of the ENS, have been recognized as potential key players in the ENS-to-CNS interaction. These cells share similarities with astrocytes from the CNS and are known for their role in maintaining neuronal health and modulating synaptic activity in the gut [64].

The ENS comprises two plexuses. The outer plexus forms the myenteric (Auberbach’s) plexus, which is situated between two muscles of the intestine. The inner ring, also called the submucosal (Meissner’s) plexus, lies between a muscle and the mucosa [63]. Both receive signals from EGCs and neurons, interacting with enteroendocrine sensory cells (EECs) that respond to environmental stimuli within the lumen [65]. In addition, enteric neurons are essential for the expression of tight junction proteins for intestinal epithelial barrier function, which is highly dependent on cytokines and thus inflammation [65].

Since approximately two-thirds of MS patients suffer from digestive dysfunction [66], interactions between the ENS and CNS were suspected early on. This is supported by a study from Wunsch et al. [67] demonstrating that degradation of the ENS occurs prior to CNS pathology in an EAE model. This deterioration was mediated by autoantibodies against antigens derived from enteric glia and neurons, leading to neuronal loss and gliosis. Overall, these findings underscore the intimate connection between the CNS and ENS.

This intricate interplay between the CNS and ENS becomes even more evident when considering the role of the vagus nerve, the 10th cranial nerve, which, despite its inability to directly receive signals from the microbiome, serves as a crucial conduit for the transmission of microbial signals indirectly sensed through the diffusion of compounds or with the assistance of other cells, such as EECs, which perceive the luminal environment and then relay the signal to the vagal nerve fibers [68]. Emerging from the abdomen, these fibers pass through the thorax and the neck to innervate the pharynx and larynx, and they ultimately reach the medulla oblongata, establishing the vagus nerve as a vital conduit for afferent signals. These signals, conveyed through the nucleus tractus solitarii (NTS), impact multiple cortical regions of the brain, such as the hypothalamus and amygdala, giving rise to a wide variety of nuanced responses that include emotional and behavioral reactions [68,69].

Notably, the connection of the ENS to the vagus nerve is bidirectional, enabling the adaptation of ENS functions to the current physiological state of the organism, ensuring the maintenance of homeostasis.

In addition, the vagus nerve has been shown to play a regulatory role in the context of inflammation. Inflammatory activation of the vagus nerve leads to the release of acetylcholine (ACh) in the gut and other organs, which binds to ACh receptors (AChRs) expressed by a wide variety of adaptive and innate immune cells [70]. For instance, signaling via the α7 nicotinic acetylcholine receptor (α7nAChR) inhibits NF-κB signaling and induces anti-inflammatory JAK2-STAT3 signaling in intestinal macrophages [71]. This inflammatory reflex orchestrated by the vagus nerve serves as a protective mechanism against proinflammatory cytokines. Previous studies have demonstrated that signaling via the α7nAChR significantly diminishes the secretion of proinflammatory cytokines by Th1 and Th17 cells [72], emphasizing the pivotal role of the vagus nerve in modulating immune responses.

While the direct contribution of astrocyte-like EGCs in MS remains to be studied in more detail, insights from gastrointestinal and neurological disorders, like PD, indicates that these cells partake in the propagation of inflammatory signals relevant for disease progression [73]. Moreover, recent evidence suggests that EGCs themselves may become peripheral targets in the course of MS progression. In patients with MS, there is a notable increase in GFAP immunoreactivity within the myenteric ganglia indicative of gliosis [67]. While clear evidence of heightened EGC reactivity during MS is lacking [74], two studies have identified circulating autoantibodies against enteric neurons and glia cells in MS and EAE [67,74]. These findings underscore the potential involvement of enteric glial cells as targets of autoimmune attack in MS, highlighting the intricate interplay between the enteric nervous system and the pathophysiology of CNS disorders.

### 3.2. Hypothalamus–Pituitary–Adrenal Axis

Another important connection between the CNS and the gut is through the neuroendocrine HPA axis (Figure 1). Activated by both internal and external stressors, this axis employs hormonal pathways to counteract real or perceived threats to the body’s homeostasis [75,76].

In response to stress signals, neurons in the paraventricular nucleus (PVN) of the hypothalamus become active and secrete corticotropin-releasing hormone (CRH). Together with other proteins, such as vasopressin, CRH migrates via the hypophysial portal vessels to the anterior pituitary. Here, CRH binds to corticotropin-releasing hormone receptor 1 (CRHR1), while vasopressin binds to the arginine vasopressin 1b receptor (Avp1b) [75,76]. This leads to the direct release of adrenocorticotropic hormone (ACTH) into the systemic circulation, where it migrates to the adrenal cortex. Within the zona fasciculata, adrenocorticotropic hormone (ACTH) binds to melanocortin receptor 2 (MC2R), leading to an increase in intracellular cyclic adenosine monophosphate (cAMP) levels. This cascade ultimately stimulates the synthesis of the glucocorticoids (GC) cortisol and corticosterone in humans and rodents, respectively [75,76], regulating a diverse range of short-term and long-term stress responses throughout the body.

Evidence suggests that acute or chronic stress can not only impact neurons but may also affect the morphology and function of various types of glial cells, including astrocytes. Several studies have reported alterations in astrocyte reactivity in response to stress [77,78,79]. Moreover, a study by Zhao et al. [80] describes a decrease in glycogen content as a potential mechanism underlying structural and molecular changes in astrocytes due to chronic stress, challenging the glucocorticoid-centered theory of stress. Conversely, exposure of astrocytes to high glucocorticoid levels can lead to enhanced expression of neurogenesis-related genes, thereby promoting adult neurogenesis [81].

Like the intricate interplay between the ENS, vagus nerve, and CNS, the microbiome, stress, and HPA axis are closely intertwined. While this interconnection is particularly pronounced during early life, stress-induced dysbiosis has the potential to disrupt the microbiome throughout adulthood, triggering aberrant immune responses [82]. In addition, the microbiome significantly influences the HPA axis, shaping the response of peripheral tissues to stress [83]. Dysfunction of this bidirectional interconnection between the HPA axis and the microbiome is linked to various disease pathologies and, notably, seems to correlate with neuroinflammation in MS patients [84]. An in-depth comprehension of the intricate interactions between stress, the microbiome, and the HPA axis is essential for unraveling their collective impact on health and disease.

### 3.3. The Blood Circulatory System and the Blood–Brain Barrier

The microbiome has the capacity to produce a vast variety of interorgan signals, including microbial components, metabolites, and neurotransmitters. Several of these signals are released into the circulation and directly act within the CNS. This makes the blood circulatory system an important pathway for interorgan communication between the intestine and CNS (Figure 1) [85].

The vascularization of the intestine extends to the epithelial cell layer. Together with the overlying mucosal layer, and the underlying lamina propia, housing immune cells, they form the different layers within the intestinal barrier and thereby provide essential metabolic function. Its main purpose is to prevent harmful substances from entering the bloodstream while facilitating the absorption of microbiome-derived products, such as nutrients, proteins, or fatty acids, which are absorbed and transported into the blood across the epithelial cell layer [86]. This transport can be active via transporters or passive by diffusion. Moreover, the semipermeable nature of the mucosa limits the transport of potentially harmful components, thereby contributing to intestinal integrity and homeostasis [87]. Apart from the mucosa, this barrier homeostasis is mainly achieved by the epithelial monolayer and its intercellular tight junctions, which are essential for regulated absorption, as they allow water and ions to pass between the epithelial cells, whilst preventing microbes or larger molecules from crossing, thus offering protection against infection [88]. Dysfunction of the intestinal barrier is also referred to as “leaky gut” and is associated with numerous diseases, including inflammatory bowel disease, AD, and MS [89]. It is conceivable that these disorders are at least partially mediated by the unhindered release of microbial mediators into the circulation, leading to aberrant responses in both the periphery and CNS.

Once in circulation, microbial signals face the challenge of crossing the BBB to reach the CNS. The BBB regulates the microenvironment and is essential for the protection of the CNS against pathogens from the periphery. It consists of endothelial cells, astrocytes, and pericytes [90]. Like those of the intestinal barrier, the interstitial space of endothelial cells is tightly sealed by tight junctions, adherent junctions, and junctional adhesion molecules [90]. In addition, astrocytic end-feet ensheathe the endothelium to enhance barrier functions and to mediate the selective entry of nutrients into the CNS [91]. Any disruption in the integrity of the intestinal barrier or BBB can therefore result in the uncontrolled release of microbial mediators. This breakdown allows potentially harmful substances originating from the gut or systemic circulation to enter tissues and trigger inflammatory responses. Astrocytes are therefore strategically located as first responders to microbial signals, with the capacity to respond by mounting inflammatory responses through the release of various signaling molecules, including cytokines, chemokines, and reactive oxygen species [92].

Interestingly, these inflammatory processes lead to increased permeability of the intestinal vascular barrier, causing the aggravation and spread of inflammation. However, in the brain, such inflammation results in closure of the choroid plexus (CP) [93].

The CP is a complex vascular network of the brain separating the CSF from the blood, which regulates inflammatory processes in the CNS. Excitingly, it has been shown that patients with neurological disorders such as MS display an altered CP volume which could potentially influence disease pathology and provide a promising target in gut–brain communication [94]; however, more research is required. In summary, the breakdown of both the intestinal and CNS barriers contributes to an imbalance that can potentially induce and exacerbate inflammatory disorders.

## 4. Mediators of Interorgan Gut–Brain Signaling

Bidirectional interorgan communication between the CNS and the gut is mediated through a wide plethora of signals of varying classes and origins, and a comprehensive discussion of their nature is beyond the scope of this review. However, while many signals have yet to be identified, they can be broadly grouped into mediators that originate from bacteria directly and signals that are the consequence of responses triggered by bacteria, influencing immune responses and BBB integrity and therefore also providing therapeutic potential (Figure 2).

### 4.1. Bacterial Metabolites

Intestinal bacteria produce a wide variety of metabolic products that are vital for the host and have a significant impact on immunity and overall health. Among these, recent research has implicated SCFAs and secondary bile acids (BAs) as important mediators of inflammatory functions in MS [95].

SCFAs are produced from dietary fiber by anaerobic bacteria through saccharolytic fermentation. In addition to their important functions during homeostasis, SCFAs strongly modulate the immune response by binding to free fatty acid (FFA) receptors 2 and 3 and the G-protein-coupled receptor 109A (GPR109A) [96]. In intestinal epithelial cells, receptor activation by SCFAs leads to the induction of NLRP3 and IL-18, thereby strengthening epithelial barrier function and boosting pathogen defense mechanisms [96,97]. Along with FFA3, FFA2 is expressed mainly on immune cells, and its activation leads to the downregulation of NF-κB, reducing the release of proinflammatory cytokines such as IL-6 and IL-1β [96]. Butyrate reduces the production of proinflammatory molecules such as iNOS, TNF-α, MCP-1, IL-12, and IL-6 [96,98] and has previously been associated with beneficial effects on the pathogenesis of MS [99,100,101]. Depending on their differing degrees of water solubility, certain SCFAs can also be absorbed into the bloodstream and enter the CNS. Here, they can directly decrease microglial activation through histone deacetylase inhibition, attenuating inflammatory responses, and providing neuroprotection [102,103,104]. This reduction in microglial activation can furthermore attenuate astrocyte reactivity, potentiating tissue-protective processes [92,105]. Overall, MS patients exhibit reduced serum SCFA levels, highlighting the relationship between the gut microbiota and autoimmune neuroinflammation [95,103]. The resulting deficiency in SCFAs may compromise the protective effects that these microbial metabolites exert on immune regulation, thereby fueling inflammation in a reciprocal feedback loop and exacerbating MS pathology.

In addition to SCFAs, BAs represent another important microbial mediator of interorgan communication between the gut and the CNS. BAs are important products of cholesterol metabolism and are involved in fat digestion and absorption [106]. Its synthesis is a multistep process and involves the synthesis of cholesterol in the liver before it is stored in the gallbladder. Upon dietary consumption, these peptides are released into the small intestine, where they not only aid digestion but also exert antimicrobial effects via farnesoid X receptor (FXR)-induced antimicrobial peptides that protect the intestine from bacterial invasion and overgrowth [107]. In the small intestine, bacteria convert primary BAs into secondary BAs, which can be further metabolized, resulting in a diverse pool of BAs in the intestinal lumen [103,107]. Primary BAs, such as cholic acid and chenodeoxycholic acid, and secondary BAs, such as lithocholic acid and ursodeoxycholic acid, enter the blood through the intestine [106,107]. While the majority of microbial metabolites are recycled in the liver, a small fraction migrate via the bloodstream through the BBB into the CNS, where they exert direct immunomodulatory and neuroprotective functions in the context of autoimmune neuroinflammation [108]. Here, BA can bind to monocytes and microglia through G protein-coupled BA receptor 1 (GPBAR1) and decrease MHC-II and CD80 expression [109] (Figure 2). Furthermore, hydrophilic ursodeoxycholic acid (UDCA), particularly tauroursodeoxycholic acid (TUDCA), has the potential to induce anti-inflammatory signaling in microglia and to prevent neurotoxic polarization of astrocytes [110]. Bhargava and colleagues [110] demonstrated that treatment of astrocytes with TUDCA blocked the upregulation of proinflammatory gene transcripts in a dose-dependent manner. Interestingly, an agonist of GPBAR1 partially replicated these effects. Additionally, astrocyte-conditioned media from TUDCA-treated cells showed reduced toxicity to murine oligodendrocytes compared to vehicle conditions, overall suggesting that TUDCA can prevent astrocyte polarization to a neurotoxic phenotype, offering potential therapeutic benefits in neuroinflammatory and neurodegenerative disorders.

In addition to their direct effects on the CNS, BAs also regulate the inflammatory functions of innate and adaptive immune cells. Binding to GPBAR1 on circulating and CNS-infiltrating monocytes reduces CD40, CD80, and MHCII expression, thereby reducing the general infiltration of immune cells into the CNS [109] (Figure 2). Moreover, some BAs can suppress the differentiation of Th17 cells and induce Treg differentiation in the intestine [111], significantly contributing to the host immune response.

Notably, several studies indicate that BA metabolism is overall reduced in MS patients. In the study by Bhargava et al. [110], the authors demonstrated that primary bile acid serum levels were significantly reduced in both primary and secondary MS patients, while secondary bile acid levels were reduced in RRMS patients. Another study by Crick et al. [112] showed that a particular bile acid, 25-hydroxycholesterol, was decreased in the serum but increased in the CSF. Moreover, it was further demonstrated that the expression of the bile acid receptor farnesoid X receptor (FXR), which typically confers anti-inflammatory effects, on peripheral immune cells is reduced in RRMS patients [113]. Consequently, alterations in SCFAs and BAs associated with anti-inflammatory properties exacerbate MS pathology by promoting a proinflammatory state. These effects may at least partially be mediated through astrocytes, as the lack of SCFAs and BAs can drive the polarization of astrocytes into pathological phenotypes. Understanding and therapeutically targeting these alterations in SCFAs and BAs may hold promise for mitigating the inflammatory processes underlying MS.

### 4.2. Bacterial Membrane Compounds

In addition to bacterial metabolites, microbial membrane components such as lipopolysaccharide (LPS), a component of the bacterial cell wall of Gram-negative bacteria, also function as interorgan signaling molecules via direct and indirect mechanisms in the CNS. Bacteria can release LPS during growth and upon death, such as after antibiotic treatment, leading to the modulation of inflammatory responses [114]. LPS is sensed by a variety of cell types via its cognate receptor TLR4. In the gut, intestinal EECs respond to LPS through the secretion of proinflammatory cytokines such as IL-6, IL-1β, and TNF [115] (Figure 2). These cytokines stimulate the afferent fibers of the vagus nerve and subsequently transmit information to the CNS [116]. LPS can also act via the HPA axis by TLR4-mediated cytokine release, leading to an elevated release of ACTH and, consequently, increased production of glucocorticoids [117]. When the intestinal barrier is compromised, the translocation of LPS into the circulation can trigger further inflammatory responses. A compromised BBB can also lead to LPS-induced inflammation in the CNS. Elevated LPS levels can further destabilize BBB integrity and activate CNS-resident cell types, such as microglia, which in turn polarize astrocytes into a proinflammatory state [20,118,119]. While the question whether astrocytes are capable of sensing LPS through TLR4 remains a topic of ongoing debate, these findings underscore the central role of astrocytes in orchestrating the CNS response to pathogen-derived signals and highlight their involvement in the detection and modulation of microbial threats within the CNS.

Recently, extracellular vesicles (EVs) are gaining increasing interest in the context of inflammatory diseases. EVs, particularly outer membrane vesicles (OMVs), have emerged as intriguing messengers between bacteria and their host, operating as part of an alternative horizontal communication system alongside direct host–microbe interactions [47,120]. OMVs, which originate from the outer membrane of bacteria, are proteolipid vesicles enriched with proteins, lipids, nucleic acids, metabolites, and virulence factors. These vesicles, which are released by both pathogenic and commensal bacteria, can enter the bloodstream, potentially cross the BBB and influence various CNS-resident cell types, including astrocytes and microglia [47,120]. Given their described association with neurodegenerative diseases, such as AD, PD, or dementia, where they drive inflammatory mechanisms, it is highly plausible that OMVs also play a significant role in the context of MS [120].

This notion is further supported by the emerging concept that EVs play important roles in MS pathology, with reports indicating their increased levels in MS patients compared to healthy controls [47,121]. These EVs have been implicated in various key processes underlying MS development, including the compromising of BBB functions, activation of immune cells during relapses, facilitation of migration across the BBB, and propagation of inflammation within the CNS. The ability of EVs to traverse the BBB has recently been demonstrated by several studies, including one by Bittel et al. [122] and others [123,124], in which the authors reveal the transfer of gut bacteria-derived BEVs to various host organs, including the CNS in murine models.

However, the functional relevance of OMVs in MS remains to be investigated and is furthermore complicated by the fact that the specific impact of OMVs on disease pathogenesis is dependent on their content and bacterial origin, with some vesicles demonstrating proinflammatory effects and others exhibiting anti-inflammatory effects [47]. It will therefore be of utmost importance to discern between disease-driving OMVs and their origin, and disease-limiting OMVs, in order to establish valuable therapeutic insights. The interaction of OMVs with astrocytes may, in this context, be of particular relevance as these cells are strategically positioned to directly engage with OMVs following BBB transgression [90]. Moreover, astrocytes have been shown to actively perform endocytosis and phagocytosis, thereby facilitating their access to the contents of OMVs [125,126,127]. Overall, this highlights the importance of studying OMVs in the context of MS. While certain OMVs may contribute to this disease, their unique properties also offer opportunities for therapeutic intervention to halt or slow disease progression. Therefore, OMVs are an intriguing subject of research and open up new avenues for therapeutic intervention in acute and chronic MS.

### 4.3. Neurotransmitters

Neurotransmitters are key signaling molecules. In the context of MS, the neurotransmitter serotonin and its precursor tryptophan have gained increasing attention due to their microbial dependence. The importance of the commensal microbiota in the production of serotonin has convincingly been demonstrated by studies using antibiotic-mediated depletion of intestinal bacteria, which resulted in significant changes in tryptophan availability and serotonin levels [128]. Indeed, tryptophan is an essential amino acid and must be obtained through our diet [129,130]. Its transformation into serotonin requires the action of bacterial enzymes within the intestine, and the process can be divided into two distinct pathways, resulting in the production of either kynurenine or serotonin (Figure 3). The degradation of tryptophan via the kynurenine pathway occurs through the rate-limiting enzyme indolamine-2,3-dioxygenase (IDO), which is distributed throughout the brain, liver, and intestine and is induced by proinflammatory cytokines (Figure 3) [129]. Among other important products of the kynurenine pathway, such as NAD^+^, the metabolites kynurenine, kynurenic acid, and xanthurenic acid are particularly relevant in the context of MS, as they serve as agonists of the aryl hydrocarbon receptor (AhR), a ligand-activated transcription factor with essential functions in immune regulation [131,132]. Indeed, AhR signaling has been demonstrated to have far-reaching immunomodulatory functions not only in peripheral organs but also in the CNS, where it controls the inflammatory properties of astrocytes and microglia [45,133,134,135,136]. In this context, we and others have observed that type-I interferons (IFN-Is) stimulate the expression of AhR in astrocytes, triggering AhR-dependent anti-inflammatory responses [134]. Intriguingly, conditional deletion of AhR in astrocytes exacerbated disease progression in EAE studies, correlating with increased production of proinflammatory mediators and increased NF-κB activation [134]. Moreover, analysis of lesion tissue from MS patients revealed upregulation of AhR in astrocytes, coinciding with activation of the IFN-I pathway [134]. Recently, we have described that AhR activation in astrocytes induces the expression of the immune checkpoint protein PD-L1, thereby attenuating inflammatory responses in both infiltrating cell types and CNS-resident microglia [15]. These findings underscore the pivotal role of AhR signaling in astrocytes as a crucial mediator of CNS inflammation and offer potential therapeutic avenues for MS and other neuroinflammatory disorders.

In addition to the role of AhR signaling in astrocytes in MS, activation of this ligand-activated transcription factor has been associated with protective and immunomodulatory functions, for instance, by increasing IL-10 expression in B cells [137] or stimulating the differentiation of regulatory T cells [138]. Collectively, several studies have demonstrated that the induction of AhR signaling has the capacity to attenuate autoimmune CNS inflammation, suggesting that AhR is an important therapeutic target that exemplifies the complexity of gut–brain interactions [139,140]. Even if these derivatives exert mostly anti-inflammatory effects through signaling via AhR, it is noteworthy that they can also exhibit proinflammatory properties. Kynurenic acid, for instance, regulates neuronal excitability and plasticity and serves as an inhibitor of the α7 nicotinic acetylcholine receptor, whose signaling is involved in the cholinergic anti-inflammatory pathway [141]. This complexity may indicate that the net ratio of the various molecules is critical for an overall favorable or adverse effect upon inflammation. In addition to the kynurenine pathway, tryptophan can be metabolized by the serotonin pathway (Figure 3), which is mainly controlled by the enzyme tryptophan hydroxylase (TPH) and results in the production of serotonin, a neurotransmitter that modulates mood, sleep, neuronal plasticity, and many gastrointestinal functions. Approximately 90% of serotonin can be found in the intestine and is synthesized by enterochromaffin cells, which are strongly regulated by the intestinal flora [142,143]. In the periphery, serotonin has the capacity to control a wide spectrum of immune-related functions in numerous cell types. For instance, activation of the 5-hydroxytryptamine (HT)_2B_ and 5-HT_7_ receptors by serotonin leads to polarization of macrophages toward an anti-inflammatory M2 phenotype [144]. Moreover, 5-HT receptor activation in lymphocytes drives IL-10 production and decreases the secretion of proinflammatory cytokines by Th1 and Th17 cells [145], suggesting the protective effects of serotonin. Furthermore, vagal afferent fibers express 5-HT receptors, indicating that serotonin may directly activate the vagus nerve and transmit signals to the CNS [146].

In the context of MS, several studies point to the relevance of serotonin in regulating inflammatory responses [147,148,149,150,151], which is likely due to alterations in tryptophan availability [135,136]. Interestingly, the abundance of AhR agonists present in the serum undergoes dynamic changes as MS progresses, highlighting the potential of AhR as a disease-stage-associated marker.

In addition to the kynurenine and serotonin pathways, tryptophan can be metabolized via the indole pathway into a variety of indoles, such as indole-3-acetic acid (IAA) and indole-3-pyruvic acid (IPA) (Figure 3) [129]. Notably, these indoles can also act as ligands for AhR and regulate inflammatory functions in a wide variety of cell types. For instance, AhR activation through indoles can regulate the differentiation and function of immune cells, such as Tregs, and drive both pro- and anti-inflammatory functions [152]. Therefore, indoles share similar immunomodulatory properties with metabolites of the kynurenine and serotonin pathways, strengthening the importance of bacterial metabolites in the regulation of inflammation.

### 4.4. Gut Hormones

In addition to neurotransmitters, hormones produced in the gut can act as important interorgan signals that modulate a plethora of immune-related functions both in the periphery and in the CNS. In these lines, the microbiota-derived hormones melatonin, glucagon-like peptide 1 (GLP1), and glucocorticoids (GCs) have received increasing attention because of their relevance to immune modulation (Figure 1).

As previously discussed, Trp is a central mediator of gut-derived inflammatory signaling. After tryptophan is metabolized into serotonin, it can be further converted to melatonin (Figure 3) [153]. Melatonin acts in turn as a regulator of the kynurenine pathway by influencing the expression of the enzyme IDO1, ultimately increasing the production of kynurenic acid [154]. Moreover, melatonin is reported to have anti-inflammatory effects by binding to AhR [155]. Additional inflammatory functions include antioxidant effects, downregulation of iNOS, and inhibition of NF-κB [156], suggesting a plausible link between melatonin and MS. Indeed, Sandyk and Awerbuch [157] showed that MS patients with a disease duration of more than 5 years have a significant reduction in melatonin. A separate study by Farez et al. [158] showed an inverse correlation between melatonin levels and clinical disease activity in MS, which is partially mediated by melatonin-driven suppression of Th17 differentiation and oxidative stress [159,160]. Moreover, the authors demonstrated that MS relapses mainly occur in spring or summer, when melatonin levels are naturally at their lowest. Tan et al. [161] hypothesized that the decrease in melatonin in MS patients is due to the deposition of calcium in the pineal gland, while others suggested aberrant hypothalamic regulation of melatonin release [162]. Collectively, these studies underscore the crucial role of melatonin in the pathology of MS.

In addition to melatonin, GLP-1 produced by gut enteroendocrine cells in response to food intake or bacterial metabolites, including indoles and S-eqol, regulates various functions in the CNS [163]. While GLP-1 produced in the gut can be taken up into the circulation and migrate to the CNS, it can also be directly produced by microglia [164]. Hormones are thought to promote neuronal survival and neurogenesis, regulate synaptic plasticity, and suppress inflammation [165,166]. The vagus nerve also expresses receptors for GLP-1, allowing the hormone to act directly on vagus fibers [163]. Moreover, it has been demonstrated that administration of GLP-1 into the brain can increase ACTH and glucocorticoid levels, indicating that it can also activate the HPA axis [167].

GCs are another group of gut-derived interorgan signals that are locally produced by intestinal crypts in response to stress or immune activation. GCs interact with two types of receptors: the low-affinity glucocorticoid receptor (GR) and the high-affinity mineralocorticoid receptor (MR) [168]. Glucocorticoids can be converted into their inactive form cortisone by 11ß-hydroxysteroid dehydrogenase (11ß-HSD) 2, which allows MR to bind aldosterone, especially in the kidney and colon [168]. However, cortisone can further be processed into active glucocorticoids by 11ß-HSD1, which may result in the local accumulation of active glucocorticoids that interact with GRs [169]. In rodents, it has already been demonstrated that the distribution of both receptors within the CNS differs. While MR is found in limbic areas such as the hippocampus, GR seems to be abundant throughout the CNS [170]. Upon ligand binding, both receptors form homodimers and translocate into the nucleus, where they bind to glucocorticoid-responsive genes and regulate gene expression [171]. Several studies have demonstrated that GCs can suppress proinflammatory signaling and promote protective functions [172,173,174]. In these lines, several studies have provided evidence that GR activation has pleiotropic functions on glial cells [175,176]. While GR stimulation in astrocytes in the context of PD drives the expression of proinflammatory transcripts [176], it can limit inflammatory signaling in microglia [175].

Moreover, downstream signaling in GCs reduces the expression of costimulatory molecules by DCs [177] and attenuates proinflammatory Th1 responses [178], collectively contributing to the resolution of inflammation. Indeed, GCs represent one of the most established therapies for acute MS relapses. High-dose methylprednisolone is commonly used to provide immediate symptomatic relief following an inflammatory episode [179]. This therapeutic approach hinges on the anti-inflammatory effect of GCs on T cells, which is mediated by the downregulation of proinflammatory mediators, such as NO, and adhesion molecules that are essential for crossing the BBB [180,181]. However, corticosteroid resistance poses an increasing challenge, and there is evidence that GCs not only suppress but also drive inflammation, particularly in the context of chronic stress [182,183]. Here, GCs fuel LPS-induced NF-κB signaling and the associated expression of proinflammatory genes in the CNS [184,185]. In general, the microbiome contributes significantly to the systemic effects of GCs [186], and altered GC levels and their associations with disease severity and progression have been reported in MS patients [187,188]. While GCs do not necessarily alter the overall course of the disease, they remain essential for alleviating acute relapses, highlighting their importance in the context of autoimmune neuroinflammation.

### 4.5. Neuroinflammation and Blood–Brain Barrier Integrity

The breakdown of the BBB occurs early during MS pathology and is triggered by autoimmune inflammation and the infiltration of autoreactive immune cells [1]. This compromised BBB not only facilitates the unhindered infiltration of immune cells into the CNS but also enables the entry of signals from the gut, intensifying inflammatory processes. The bidirectional impact of BBB breakdown highlights its dual role in promoting both immune cell migration and the transmission of signals that exacerbate inflammation in the context of MS.

Gut-derived signals also have important functions in the regulation of barrier functions. For instance, intestinal bacteria can stimulate the release of zonulin, a molecule that normally regulates tight junctions in the gut [189]. In MS patients with active disease, elevated zonulin levels can increase BBB permeability and may also contribute to a leaky gut, allowing bacterial metabolites and immune mediators to “leak” into the circulation and the CNS, thereby further contributing to neuroinflammatory processes [190,191]. Consequently, bacterial metabolites with proinflammatory functions can enter the CNS and drive pathological processes.

Moreover, SCFAs have been demonstrated to fulfill important functions in BBB regulation, as the addition of butyrate can upregulate tight junction expression and restore barrier integrity [192]. In MS patients, SCFA levels are reduced, potentially further contributing to an impaired BBB [95,193]. Overall, many gut-derived anti-inflammatory interorgan mediators relevant to BBB regulation are decreased in MS patients, underscoring their significance in the initial infiltration of autoreactive immune cells. Moreover, microbial dysbiosis and its association with a disrupted intestinal and blood–brain barrier may play a significant role in fostering the proinflammatory environment during both the acute and chronic stages of MS. In summary, the entirety of the alterations in gut microbiome composition and the secretion of bacterial metabolites discussed in this review not only deepen our understanding of the intricate interplay between the gut and the CNS but also offer a compelling and potent avenue for advancing innovative therapeutic strategies to address the unmet medical need in MS patients.

## 5. Therapeutic Outlook

The currently available treatments for MS, including immunosuppressants, antibodies, or steroids, primarily focus on mitigating inflammation during the acute stages of MS by targeting the initial infiltration of autoreactive immune cells. While these approaches effectively limit further damage to axons and myelin and delay disease progression, they come at the expense of substantial side effects and high costs. Hence, there is growing interest in alternative or supplementary treatments. Given the robust association between dysbiosis and MS, as well as other autoimmune diseases, there is increasing emphasis on harnessing the potential of the gut microbiome as a promising target for therapeutic intervention. This innovative approach not only optimizes resource utilization but also stands out as a preventive strategy, potentially alleviating the burden of costly interventions associated with disease progression. Beyond its cost-effectiveness, therapies targeting the microbiome offer a broad spectrum of beneficial functions, extending well beyond the confines of specific diseases.

### 5.1. Probiotic Treatment

One way to affect the composition of bacteria in the intestine is via the use of probiotics. According to the FAO/WHO, probiotics are living microbes that, when administered, benefit the health of patients [194]. Among the most commonly used probiotics are Lactobacillus and Bifidobacterium, both of which produce anti-inflammatory molecules such as SCFAs and are therefore considered to have health-promoting properties [195]. These genera have also been included in several probiotic studies in MS patients and EAE models, and their effects on MS pathology and symptoms have been thoroughly investigated (Table 1).

In an EAE study, the administration of a polyprobiotic consisting of Lactobacillus casei, Lactobacillus acidophilus, Lactobacillus reuteri, Bifidobacterium bifidum, and Streptococcus thermophilus resulted in elevated numbers of Treg cells and increased secretion of the anti-inflammatory cytokine IL-10 [196]. Similar preclinical studies were conducted using Lactobacillus plantarum together with Bifidobacterium animalis, which improved neuroinflammation and increased Treg cell numbers in the lymph nodes and spleen [197]. Furthermore, treatment with Lactobacillus reuteri in an EAE model was able to reduce the number of Th1 and Th17 cells and their corresponding cytokines and restore microbiome diversity in the gut [198].

Similar effects were observed in studies involving MS patients [200,201,202]. The administration of Lactobacillus acidophilus, Lactobacillus casei, Bifidobacterium bifidum, and Lactobacillus fermentum to MS patients resulted in a decrease in proinflammatory cytokines, including IL-8 and TNF-α [201], and an improvement in disability, as measured by the expanded disability status scale (EDSS) [200]. Another study tested the probiotic supplement VSL#3 in the context of MS [202]. VSL#3 is commonly used in gastrointestinal diseases, such as IBS, and consists of eight different bacteria, all of which are Lactobacillus, Bifidobacterium, or Streptococcus. The addition of this polyprobiotic induced anti-inflammatory responses, most notably reducing the number and inflammatory functions of innate immune cells [202]. In summary, these findings collectively suggest that the genera Lactobacillus, Bifidobacterium, and Streptococcus exhibit consistent health-promoting qualities, highlighting their substantial therapeutic value in the context of MS.

With the increasing prominence of this potential therapeutic approach, numerous studies are investigating novel bacterial strains that may harbor beneficial effects on MS patients. A recent study showed a reduction in proinflammatory cytokines after the administration of *Bacillus coagulans* in EAE mice [199]. *B. coagulans* is reported to inhibit the expression of the enzyme IDO, resulting in diminished degradation of tryptophan through the kynurenine pathway, which, in turn, facilitates the increased availability of tryptophan for serotonin synthesis. This approach could prove highly advantageous because it provides the opportunity to selectively modulate a dysregulated pathway with profound implications in the context of MS. However, it is important to note that one probiotic bacterial strain may vary when administered independently compared to when it is administered in combination with other probiotic bacteria [203]. In summary, the beneficial effects of probiotic substitution need to be considered and validated in a complex environment involving multiple probiotics and the endogenous microbiome [203,204].

Despite their numerous beneficial effects, probiotics have significant limitations, including the need for consistent use to ensure long-lasting beneficial effects. Notably, probiotics may not be sufficient as a standalone treatment for acute inflammation, given that their beneficial effects are contingent on long-term adaptations rather than providing immediate intervention. Furthermore, the variable effects of probiotics, influenced by factors such as the host’s diet and environment, pose challenges in predicting and ensuring consistent positive outcomes across diverse populations. These limitations emphasize the importance of a personalized approach when considering probiotics as part of a treatment strategy in the context of MS as well as other diseases. Nevertheless, the use of probiotics represents a novel therapeutic avenue with significant potential not only for the treatment of MS but also for the treatment of other autoimmune diseases and neurological disorders.

### 5.2. Prebiotic and Herbal Supplementation

In contrast to probiotics, prebiotics are nonliving food components, including fiber and nondigestible dietary components. Prebiotics are naturally present in the diet but can also be supplemented. Examples include prebiotics such as inulin, oligofructose, and lactosucrose, which serve as crucial energy sources for bacteria in the gastrointestinal tract. In many instances, their presence leads to an augmented production of SCFAs and enhanced regulation of mucins in the gut [205,206]. Prebiotics are commonly supplemented along with probiotics, a combination approach also referred to as synbiotics [207]. This strategy involves the addition of prebiotics to probiotic preparations to increase the survival of probiotic bacteria, thereby maximizing their enduring beneficial functions [208].

While the majority of MS studies have predominantly focused on probiotics, there is a compelling rationale for considering the beneficial properties of prebiotics. By promoting the growth of beneficial bacteria, prebiotics may create an environment that supports the prolonged survival and optimal functioning of both naturally occurring commensal bacteria and externally supplemented probiotics. This synergistic approach of combining prebiotics with probiotics has the potential to positively impact the gut–brain axis and modulate the immune system, potentially offering an effective long-term strategy for managing MS.

### 5.3. Fecal Microbiota Transplantation

An alternative avenue for modulating the composition of intestinal bacteria is through fecal microbiota transplantation (FMT). This procedure involves the administration of a homogenized solution of fecal matter from a healthy donor into the intestinal tract of the recipient. The first FMT can be traced back to the 18th century in China and is currently performed as a treatment for *C. difficile* infection or inflammatory bowel disease (IBD) [209]. The therapeutic potential of this approach is increasingly being recognized in the context of other diseases associated with intestinal dysbiosis, such as MS [210,211]. In EAE models, FMT promotes a health-promoting bacterial composition, accompanied by strengthened and functional BBB integrity, reduced demyelination, and axonal loss [212]. Furthermore, in MS patients, FMT resulted in a significant improvement in motor and gastrointestinal functions, representing the first evidence of the therapeutic potential of FMTs for the treatment of MS [213,214].

Overall, FMT shows promise for restoring the commensal microbiome composition, fostering a diverse and balanced microbial community. Consequently, FMT may contribute to health-promoting interorgan crosstalk, influencing systemic functions beyond the gut. Additionally, FMT has been shown to restore intestinal permeability, preventing the translocation of harmful substances in the context of MS [213].

Despite these promising observations, the transient nature of FMT has led to concerns about its long-term efficacy, necessitating repeated treatments [215]. Logistical challenges in fecal material storage may impact microbiota viability, and the lack of standardized protocols contributes to variability in outcomes and limits reproducibility. Moreover, despite donor screening, the risk of pathogen transfer compromises the safety and sustainability of conventional FMT. Ongoing research is crucial to address these limitations, improve safety, and enhance the long-term effectiveness of FMT as a therapeutic approach. Nevertheless, the multifaceted effects of FMT underscore its potential as a therapeutic intervention with broad implications for gut and systemic health.

### 5.4. Alternative Therapeutic Approaches

Research into alternative therapeutic approaches for MS has led to the investigation of direct administration of bacterial-derived metabolites such as BA. In particular, TUDCA has shown considerable promise in preventing the polarization of astrocytes and microglia toward neurotoxic phenotypes [110]. Moreover, a recent study by Ho et al. [216] investigated the use of BA receptor agonists and demonstrated the potential of the synthetic BA obeticholic acid for attenuating disease severity in an EAE model. These findings suggest that targeting the BA signaling pathway may have substantial therapeutic potential for the treatment of MS.

In addition to the administration of BA, additional therapeutic approaches using or altering bacterial metabolites in the context of MS include the administration of anti-inflammatory SCFAs. Erny and colleagues [217] showed that supplementation with an SCFA mixture with drinking water in GF mice promoted anti-inflammatory microglial functions. Indeed, SCFA supplementation has proven effective at reducing clinical severity and inflammation in EAE models, whereas the impact of high-fiber diet approaches on MS has yet to be validated [191]. Notably, the SCFA propionic acid appears to be particularly promising, as Duscha et al. [218] observed a reduction in the number of Th17 cells as well as an increase in the number and regulatory functions of Tregs upon propionic acid exposure. These results suggest that the regulation of pathologically relevant T cells by microbial metabolites may be a promising avenue for the treatment of MS.

Nevertheless, several limitations associated with the administration of BAs and SCFAs need to be considered. One limitation is the complexity of the gut microbiota, as individual responses to these interventions can vary based on the composition of an individual’s microbiome. Moreover, the optimal dosage and duration of administration for bile acids and SCFAs in MS patients are not yet well defined, and achieving reproducible effects can be challenging. The administration of bacterial metabolites nevertheless holds significant potential in MS treatment, as demonstrated by their ability to modulate immune responses and influence disease outcome.

As the quest for novel therapeutics in the context of MS has increased prominence, OMVs are emerging as an innovative therapeutic approach with substantial potential. This is primarily fueled by the ability to modify OMVs, transforming them into effective drug delivery vehicles that can be loaded with therapeutics for targeted treatments [219,220]. In addition to engineered OMVs, which hold immense potential as therapeutic agents, naturally occurring OMVs also exhibit favorable properties that can provide beneficial effects in the context of MS. For instance, OMVs produced by Bacteroides fragilis, a bacterium with protective effects in MS, are sensed by DCs via TLR2, leading to an increase in the number of Tregs and the production of anti-inflammatory cytokines [221]. In addition, OMVs derived from *Akkermansia muciniphila* have been shown to induce serotonin secretion in the colon and hippocampus [222]. While further studies will be necessary to comprehensively establish OMVs as therapeutic targets, these results provide the first evidence of the versatile therapeutic potential of OMVs in the context of MS.

Overall, modulating interorgan crosstalk between the gut and CNS represents a new frontier of therapeutic approaches that has the potential to significantly advance the treatment of MS. Preventative strategies may enable the control of disease processes during early stages, thereby reducing the risk of progression and chronification. Future studies will have to investigate how we can harness the beneficial functions of a healthy commensal microbiome to specifically modulate distinct processes of the disease, irrespective of environmental or dietary variations. These insights will lay the foundation for the development of novel therapeutics that may be effective not only for the treatment of MS but also for the treatment of other autoimmune and neurological disorders.

## 6. Conclusions

Taken together, gut-derived bacterial metabolites can be grouped into secreted metabolites, membrane compounds, neurotransmitters, and hormones. All of these microbiome-derived signals are key constituents of interorgan crosstalk. In the context of gut–CNS interactions, these signals have the capacity to alter inflammatory responses—making them important regulators of MS pathology. However, at the same time, microbiome-derived signals hold the potential to attenuate neuroinflammatory processes and improve regeneration, highlighting their relevance in addressing the unmet medical need of novel therapeutic interventions for MS.

## Figures and Tables

**Figure 1 cells-13-00497-f001:**
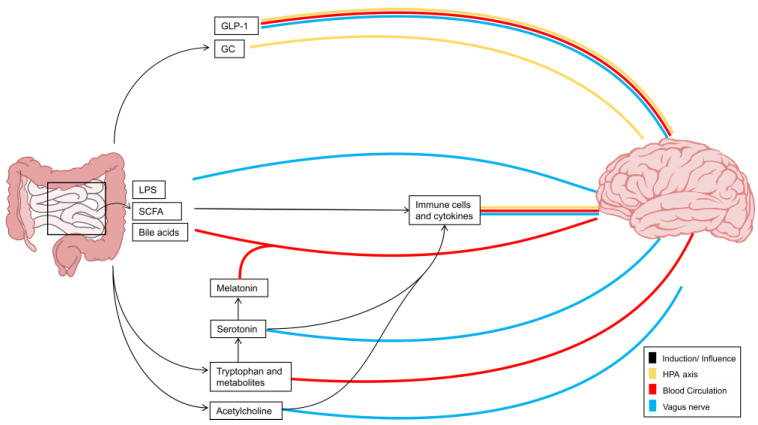
Interorgan signaling via gut–brain pathways. Interorgan communication between the CNS and the gut is implemented by a range of molecules which transmit information to the brain via several different pathways. These pathways comprise the vagus nerve, the hypothalamic–pituitary–adrenal axis, and the blood circulation. While neurotransmitters such as acetylcholine or serotonin communicate primarily via the vagus nerve, serotonin precursors and derivatives tend to travel via the bloodstream. Gut-derived LPS, SCFAs, or bile acids likewise transmit information along the vagus nerve or reach the brain via the bloodstream; however, they also interact with immune cells and thus influence the release of cytokines, thereby providing an additional indirect communication pathway between the gut and the brain. Intestinal hormones such as GLP1 and GC additionally signal via the HPA axis. Black arrows illustrate the induction or influence of the signals on each other. The blood circulation is shown in red, the vagus nerve in blue, and the HPA axis in yellow. GC: glucocorticoids, GLP-1: glucagon-like peptide-1, HPA: hypothalamic–pituitary–adrenal axis, LPS: lipopolysaccharide, SCFA: short-chain fatty acid.

**Figure 2 cells-13-00497-f002:**
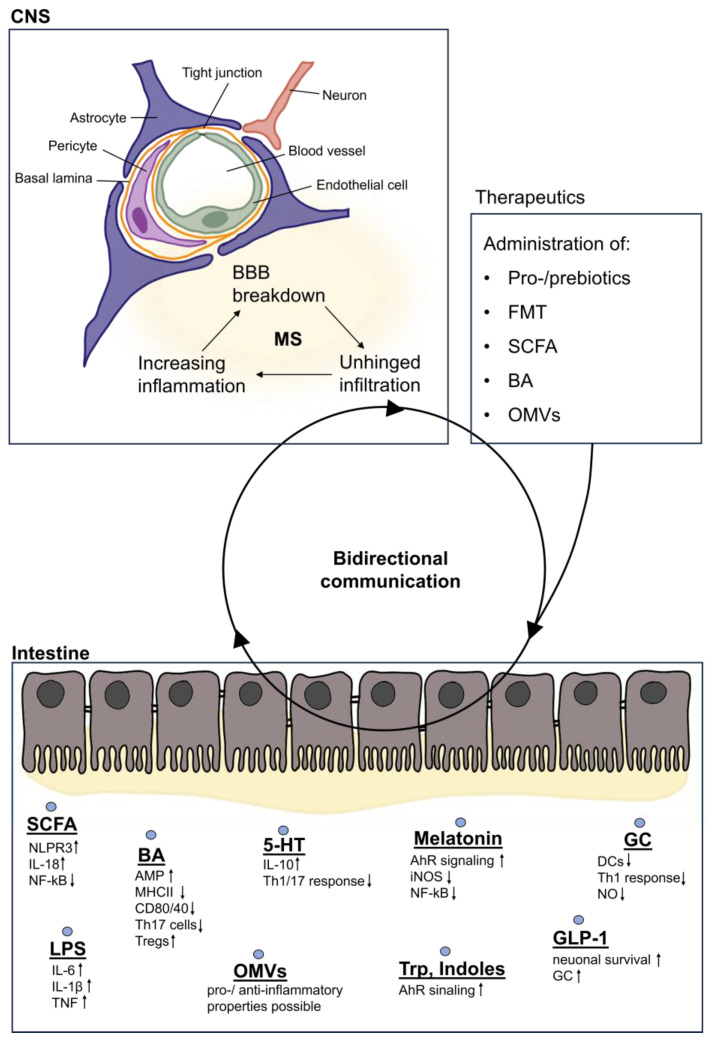
Gut-derived metabolites involved in bidirectional communication. In the intestine, metabolites are either secreted and further processed or are part of the bacterial membrane. Secreted SCFAs lead to increased production of NLPR3 and IL-18, which promotes the intestinal barrier integrity, while proinflammatory signaling pathways such as NF-kB are dampened. Similarly, BAs attenuate inflammatory responses by reducing the expression of receptors such as MHCII, CD80, or CD40. On the other hand, LPS contained in bacterial membranes leads to increased proinflammatory cytokine release. Derivatives of the tryptophan metabolism exert mainly anti-inflammatory properties as they are ligands of the AhR. Gut-derived hormones such as GLP-1 promote neuronal survival and GC attenuates DC and Th1 cell responses, thus providing anti-inflammatory effects. These signals can bidirectionally communicate with the CNS via different pathways of the GBA and thus influence the integrity and homeostasis of the brain. In MS, a breakdown of BBB integrity can be observed, also resulting in the unhindered infiltration of harmful metabolites, which in turn exacerbates inflammation and triggers a negative feedback loop. In order to support the integrity of the CNS and its function, today’s research focuses on attempts to alter this communication between the gut and the brain by the administration of metabolites, in order to achieve a higher ratio of beneficial metabolites which usually are lacking in MS patients. AMPs: antimicrobial peptides, BAs: bile acids, DCs: dendritic cells, FMT: fecal microbiota transplantation, GC: glucocorticoids, GLP-1: glucagon-like peptide, LPS: lipopolysaccharide, OMVs: outer membrane vesicles, SCFA: short-chain fatty acids, Trp: tryptophan, 5-HT: serotonin.

**Figure 3 cells-13-00497-f003:**
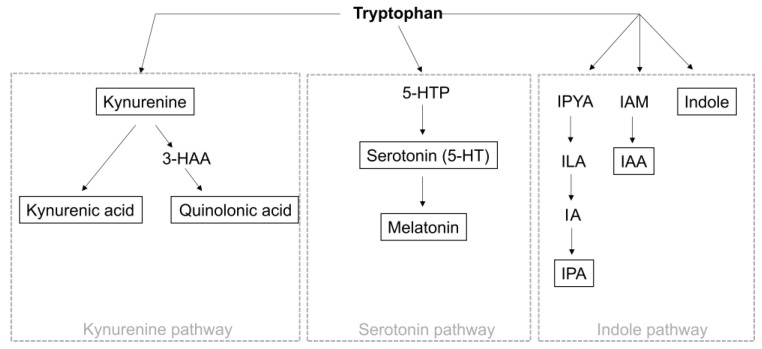
Tryptophan metabolism. Tryptophan metabolism can be divided into the indole pathway, serotonin pathway, and kynurenine pathway. Degradation via the kynurenine pathway (1) occurs via TDO and IDO, whereby tryptophan is converted to kynurenine. Kynurenine is then metabolized to kynurenic acid or via the intermediate 3 HAA to quinolonic acid. Metabolism via the serotonin pathway (2) occurs by converting tryptophan to 5-HTP by TPH1/2, which is further decarboxylated to serotonin. Serotonin can then be metabolized to melatonin. In the indole pathway (3), tryptophan is degraded to IPYA, ILA, and IA prior to IPA. Alternatively, tryptophan can be converted to indole or IAM and then to IAA. TDO: tryptophan 2,3-dioxygenase, IDO: indoleamine 2,3-dioxygenase, 3-HAA: 3-hydroxyanthranilic acid, TPH: tryptophan hydroxylase, 5-HTTP: 5-hydroxytryptophan, IPYA: indole-3-pyurvic acid, ILA: indole-3-lactic acid, IA: anholocyclic acid, IPA: indole-3-propionic acid, IAM: indole-3-acetamide, IAA: indole-3-acetic acid.

**Table 1 cells-13-00497-t001:** Probiotic bacteria and their effects upon supplementation in the context of EAE and MS.

Bacteria	Model	Observation	Reference
*Lactobacillus casei*, *Lactobacillus acidophilus*, *Lactobacillus reuteri*, *Bifidobacterium bifidum*, *Streptococcus thermophilus*	EAE	Increased Treg count Increased Il-10 level	[196]
*Lactobacillus plantarum*, *Bifidobacterium animalis*	EAE	Increased Treg count in lymph nodes and spleen	[197]
*Lactobacillus reuteri*	EAE	Reduced Th1 and Th17 count Re-establishment of microbial diversity	[198]
*Bacillus coagulans*	EAE	Reduced proinflammatory cytokines	[199]
*Lactobacillus acidophilus*, *Lactobacillus casei*, *Bifidobacterium bifidum*, *Lactobacillus fermentum*	MS	Reduced IL-8 and TNFα levels Improved EDSS score	[200,201]
VSL#3	MS	Reduced monocyte count HLA-DR downregulation by DCs	[202]

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
