# Peer review of "Gut–Brain Interactions and Their Impact on Astrocytes in the Context of Multiple Sclerosis and Beyond"

_cells, 2024, doi:10.3390/cells13060497_

Round 1

Reviewer 1 Report

Comments and Suggestions for Authors

The Review summarizes very well the role of the gut microbiota-brain axis in Multiple Sclerosis (MS). The fascinating interaction between gut microbiota and brain is first introduced from a more general point of view and then related to Multiple Sclerosis, which is also clearly introduced.

In my opinion, the manuscript is clear, well written and complete. However, it is necessary to add one or more tables summarizing the cited works and at least one/two figures representing key topics (e.g. chapter 2.2 and 4). This will facilitate the reading of this intense manuscript.

In addition, I suggest the authors to:

-         add a clearer description of the intestinal barrier in chapter 3.3.

-         briefly address the role of the choroid plexus vascular barrier in but-brain communication.

-        give a clearer explanation of OMVs route to the brain (chapter 4.2).

As minor concerns:

Line 137: Nogal et al, n.d., please insert the correct year of publication.

Figure 1: please add a more extensive description of the figure in the legend, including abbreviations.

A brief overall conclusive chapter (a few lines) at the end of the review would be appreciated.

Author Response

1.1 The Review summarizes very well the role of the gut microbiota-brain axis in Multiple Sclerosis (MS). The fascinating interaction between gut microbiota and brain is first introduced from a more general point of view and then related to Multiple Sclerosis, which is also clearly introduced.

Response: We thank the reviewer for this positive assessment.

1.2 In my opinion, the manuscript is clear, well written and complete. However, it is necessary to add one or more tables summarizing the cited works and at least one/two figures representing key topics (e.g. chapter 2.2 and 4). This will facilitate the reading of this intense manuscript.

Response: In response to the reviewer’s suggestion, we have added an additional figure addressing the main messages discussed in chapter 4.

1.3 In addition, I suggest the authors to:

  • add a clearer description of the intestinal barrier in chapter 3.3.
  • briefly address the role of the choroid plexus vascular barrier in gut-brain communication.
  • give a clearer explanation of OMVs route to the brain (chapter 4.2).

Response: We thank the reviewer for these helpful suggestions. In the updated version of the manuscript, we now provide an extended description of the intestinal barrier (lines 405-408, 413-417). We have additionally extended the section on the role of the choroid plexus vascular barrier in the context of gut-brain communication (lines 438-445). Lastly, we have provided a clearer description of how OMVs reach the brain by entering the circulation (line 583).

1.4 As minor concerns: Line 137: Nogal et al, n.d., please insert the correct year of publication.

Response: We thank the reviewer for highlighting this oversight. We have updated the respective reference in the updated version of the manuscript.

1.5 Figure 1: please add a more extensive description of the figure in the legend, including abbreviations.

Response: In response to the reviewer’s suggestion and to improve the readability of figure 1, we have significantly expanded the figure legend.

1.6 A brief overall conclusive chapter (a few lines) at the end of the review would be appreciated.

Response: In response to the reviewer’s comment, we have added a conclusion section in the updated version of the manuscript (lines 984-992).

Reviewer 2 Report

Comments and Suggestions for Authors

The review concerns an interesting and important problem of complex interactions between the gut microflora and CNS, in particular, in the context of MS and its therapy. The authors demonstrate multiple pathways involved in such interactions and suggest therapeutic approaches based on the presented evidence.

The review is well and clearly written. There are only minor comments and considerations, as follows.

1.    In lines 527-544 the authors describe the role of metabolites derived from Trp degradation pathway. In particular, they mention kynurenine, kynurenic acid and xanthurenic acid as ligands of AhR receptors mediating anti-inflammatory signaling. Here, it should be noted that kynurenic acid is also an inhibitor of α7 nicotinic acetylcholine receptors (Hilmas et al., 2001) involved in cholinergic anti-inflammatory pathway. Therefore, by affecting α7 nAChRs in the brain, kynurenic acid may exert pro-inflammatory effect.

2.    In lines 451-454 it is written that “While the majority of microbes are recycled in the liver, a small fraction migrates via the bloodstream through the BBB into the CNS, where they exert direct immunomodulatory and neuroprotective functions in the context of autoimmune neuroinflammation (Fettig and Osborne, 2021)”. Do the authors really mean that the microbes migrate to the brain or they meant the microbial metabolites?

3.    Lines 786-787: “In EAE models, FMT promotes a health promoting bacterial composition, accompanied by increased BBB permeability, reduced demyelination, and axonal loss (Li et al., 2020)”. Probably, this is a mistake, because in the paper of Li et al. it is written that “FMT led to reduced activation of microglia and astrocytes and conferred protection on the blood-brain barrier”, i.e. decreased BBB permeability.

4.    Line 816: “additional therapeutic approaches targeting bacterial metabolites in the context of MS include the administration of anti-inflammatory SCFAs”. In fact, “targeting” is not a correct term; in this context, the authors mean “using” or “applying” microbial metabolites.

5.    Minor comments.

-  line 337: last two words are spare;

-  line 512: “outer OMW”, the word “outer” is spare;

-  line 519: abbreviation MS has already been explained.

Author Response

2.1 The review concerns an interesting and important problem of complex interactions between the gut microflora and CNS, in particular, in the context of MS and its therapy. The authors demonstrate multiple pathways involved in such interactions and suggest therapeutic approaches based on the presented evidence.

The review is well and clearly written. There are only minor comments and considerations, as follows.

Response: We are grateful for the reviewer’s positive assessment.

2.2 In lines 527-544 the authors describe the role of metabolites derived from Trp degradation pathway. In particular, they mention kynurenine, kynurenic acid and xanthurenic acid as ligands of AhR receptors mediating anti-inflammatory signaling. Here, it should be noted that kynurenic acid is also an inhibitor of α7 nicotinic acetylcholine receptors (Hilmas et al., 2001) involved in cholinergic anti-inflammatory pathway. Therefore, by affecting α7 nAChRs in the brain, kynurenic acid may exert pro-inflammatory effect.

Response: We thank the reviewer for highlighting this important concept and now discuss the suggested reference in the updated version of the manuscript (lines 656-662).

2.3 In lines 451-454 it is written that “While the majority of microbes are recycled in the liver, a small fraction migrates via the bloodstream through the BBB into the CNS, where they exert direct immunomodulatory and neuroprotective functions in the context of autoimmune neuroinflammation (Fettig and Osborne, 2021)”. Do the authors really mean that the microbes migrate to the brain or they meant the microbial metabolites?

Response: We thank the reviewer for highlighting this oversight and have corrected the sentence, now referring to “microbial metabolites” instead of “microbes”.

2.4 Lines 786-787: “In EAE models, FMT promotes a health promoting bacterial composition, accompanied by increased BBB permeability, reduced demyelination, and axonal loss (Li et al., 2020)”. Probably, this is a mistake, because in the paper of Li et al. it is written that “FMT led to reduced activation of microglia and astrocytes and conferred protection on the blood-brain barrier”, i.e. decreased BBB permeability.

Response: We thank the reviewer for noting this mistake and have corrected the statement accordingly.

2.5 Line 816: “additional therapeutic approaches targeting bacterial metabolites in the context of MS include the administration of anti-inflammatory SCFAs”. In fact, “targeting” is not a correct term; in this context, the authors mean “using” or “applying” microbial metabolites.

Response: We agree with the reviewer that the use of “targeting” in this context is misleading and have therefore reworded this section according to the reviewer’s suggestion.

2.6 Minor comments.

-  line 337: last two words are spare;

-  line 512: “outer OMW”, the word “outer” is spare;

-  line 519: abbreviation MS has already been explained.

Response: We thank the reviewer for highlighting these mistakes and have corrected them accordingly in the updated version of the manuscript.

Reviewer 3 Report

Comments and Suggestions for Authors

In this review, the authors highlight the role of the gut microbiome in MS and the importance of the gut–brain axis in this pathology. They discuss the role of bacterial metabolites and hormones and present new therapeutic interventions, including pro- and prebiotic treatments.

There are several aspects that deserve the authors' attention.

1-    There are many reviews (more than a hundred) dealing with the involvement of the microbiota and the gut-brain axis in multiple sclerosis. Over seventy reports from the last three years. In addition, many of them deal with bacterial metabolites or hormones. Consequently, it is not clear what the specific contribution of this review is, apart from the fact that it presents just "another" version. The aim of the review therefore needs to be made more explicit.

2-    The title of the review is too general. It should emphasize the specifics of this review. If there are any.

3-    The expression “in the context of MS” is repeated twenty times in the review. This restricts the narrative.

4-    It is not clear why too many lines are devoted to astrocytes in the introduction (from line 90 to 107), these couple of paragraphs seem superfluous. After all, the focus of the review is on the microbiota and not on astrocytes.

5-    The sentence in lines 112-113 is very inaccurate and should be corrected. Viruses are completely different from microorganisms because they are supramolecular complexes and not living beings. Furthermore, bacteria, archaea and unicellular fungi should not be considered microorganisms. They should be considered as “microbes” because they are unicellular and have no organs in their organizational structure. Therefore, it is proposed to replace the term "microorganism" with "microbes and viral entities" to correctly capture this broader classification (in fact, microorganisms are mentioned three times and microbes four times).

6-    From line 111 (including the entire sections 2.1 and 2.2) to line 226, the information is rather general and not focused on multiple sclerosis. It can therefore be summarized very briefly instead of extending over 116 lines of text.

7-    In section 3, multiple sclerosis is only mentioned in passing (lines 284, 308 for 3.1, 371 for 3.2, 389 for 3.3). From line 278 to 405 there are a further 127 lines of text with a faint reference to MS. This section can therefore be considerably shortened without affecting the focus of the overview.

8-    A figure (schematic diagram) specifically summarizing section 4 would be useful. Figure 1 is mentioned again, but this figure lacks the involvement of bacteria.

Comments on the Quality of English Language

no comments

Author Response

In this review, the authors highlight the role of the gut microbiome in MS and the importance of the gut–brain axis in this pathology. They discuss the role of bacterial metabolites and hormones and present new therapeutic interventions, including pro- and prebiotic treatments.

There are several aspects that deserve the authors' attention.

3.1 There are many reviews (more than a hundred) dealing with the involvement of the microbiota and the gut-brain axis in multiple sclerosis. Over seventy reports from the last three years. In addition, many of them deal with bacterial metabolites or hormones. Consequently, it is not clear what the specific contribution of this review is, apart from the fact that it presents just "another" version. The aim of the review therefore needs to be made more explicit.

Response: We appreciate the reviewer's comment and acknowledge the extensive body of literature on the involvement of the gut-brain axis in neurological disorders. While it is true that the field has gained increasing attention, we believe that our review adds value by offering a comprehensive and up-to-date synthesis of the current consensus on the role of gut-brain interactions in MS. Our aim is not solely to introduce "another" version, but rather to provide a nuanced analysis that synthesizes the latest research findings and identifies emerging trends and knowledge gaps.

Furthermore, while previous reviews may have touched upon similar themes, our review distinguishes itself by transferring scientific insights in the context of gut-brain interactions to MS. To address the reviewer's concern, we have revised our manuscript to underscore this specific contribution more explicitly.

3.2 The title of the review is too general. It should emphasize the specifics of this review. If there are any.

Response: We have adjusted to title to emphasize the specifics of our review article more clearly.

3.3 The expression “in the context of MS” is repeated twenty times in the review. This restricts the narrative.

Response: In response to the reviewer’s comment, we have rephrased the respective sections.

3.4 It is not clear why too many lines are devoted to astrocytes in the introduction (from line 90 to 107), these couple of paragraphs seem superfluous. After all, the focus of the review is on the microbiota and not on astrocytes.

Response: We appreciate the reviewer's comment and the opportunity to elaborate on this aspect. Our intention was indeed to introduce novelty by extending well-established concepts of gut-brain interaction to their impact on astrocyte functions, particularly in MS. We firmly believe that this approach distinguishes our review from existing literature. To reinforce this objective, we have extensively revised the manuscript, incorporating additional references and text passages that highlight the influence of the discussed concepts on astrocyte functionality. Additionally, we have revised the title and abstract of the review to reflect these adjustments.

3.5 The sentence in lines 112-113 is very inaccurate and should be corrected. Viruses are completely different from microorganisms because they are supramolecular complexes and not living beings. Furthermore, bacteria, archaea and unicellular fungi should not be considered microorganisms. They should be considered as “microbes” because they are unicellular and have no organs in their organizational structure. Therefore, it is proposed to replace the term "microorganism" with "microbes and viral entities" to correctly capture this broader classification (in fact, microorganisms are mentioned three times and microbes four times).

Response: We thank the reviewer for highlighting this point and have made the respective corrections throughout the manuscript.

3.6 From line 111 (including the entire sections 2.1 and 2.2) to line 226, the information is rather general and not focused on multiple sclerosis. It can therefore be summarized very briefly instead of extending over 116 lines of text.

Response: We appreciate the reviewer's feedback regarding the scope and focus of our review. Our intent in sections 2.1 and 2.2 was to provide comprehensive background information on the gut-brain axis and its relevance to neurological disorders, including MS. We aimed to ensure that readers have the necessary foundation to understand the subsequent discussion on the specific role of the gut microbiome in MS pathology. However, we acknowledge the reviewer's point that these sections may appear overly generalized, and if the editorial team deems it necessary to streamline these sections, we are fully prepared to do so.

3.7 In section 3, multiple sclerosis is only mentioned in passing (lines 284, 308 for 3.1, 371 for 3.2, 389 for 3.3). From line 278 to 405 there are a further 127 lines of text with a faint reference to MS. This section can therefore be considerably shortened without affecting the focus of the overview.

Response: We thank the reviewer for pointing out these limitations. In response, we have significantly rewritten these passages in order to strengthen the relevance of gut brain interaction in MS.

3.8 A figure (schematic diagram) specifically summarizing section 4 would be useful. Figure 1 is mentioned again, but this figure lacks the involvement of bacteria.

Response: We thank the reviewer for this suggestion. In response, we have added an additional figure (Figure 2), addressing the concepts summarized in section 4 of the manuscript.

Round 2

Reviewer 1 Report

Comments and Suggestions for Authors

I am satisfied with how the authors anwered my queries.

Reviewer 3 Report

Comments and Suggestions for Authors

The authors have addressed most of the questions and comments, consequently the manuscript has been improved in precision.